# Joint modeling of alcohol and tobacco use among adults in Uganda

Grace Kakaire[ORCID][1]*, Edna Chepkemoi Chumoh[2]

1 Department of Statistical Methods and Actuarial Sciences, School of Statistics and Planning, Makerere University, Kampala, Uganda, 2 Department of Mathematics, Physics and Computing, School of Science and Aerospace Studies, Moi University, Eldoret, Kenya

* kakairegrace2@gmail.com

## Abstract

Alcohol and tobacco use are leading modifiable risk factors for non-communicable diseases (NCDs) worldwide. In sub-Saharan Africa, including Uganda, these behaviors are increasingly common but often analyzed separately, limiting understanding of their shared determinants. This study used nationally representative data from the 2014 Uganda WHO STEPS survey to examine individual and joint predictors of current alcohol and tobacco use among adults aged 18–69. Using separate and joint logistic regression models, we assessed associations with demographic, socio-economic, and health-related factors, applying a shared random effect in the joint model to account for within-individual correlation. The joint model provided better fit (AIC = 6850.13) than the combined separate models (AIC = 6890.73), suggesting a shared latent structure influences both outcomes. Female sex and higher educational attainment were consistently associated with lower odds of substance use, while older age and being separated, divorced, or widowed were linked to higher odds. Smoking-specific interactions showed stronger effects in older and underweight individuals and regional differences were more pronounced for smoking than alcohol. These findings highlight the presence of shared and distinct determinants underlying alcohol and tobacco use in Uganda. While the results suggest potential value in considering these behaviors jointly, further research is needed to evaluate whether integrated intervention strategies are effective in reducing co-occurring substance use.

## Introduction

Alcohol and tobacco use are among the top five global risk factors for morbidity and mortality, collectively responsible for more than 10 million deaths globally each year, according to recent Global Burden of Disease estimates [1,2]. These substances are not only independently harmful but are frequently co-used, often reinforcing each other's effects and complicating cessation efforts. In low- and middle-income countries

**Data availability statement:** The data underlying the results presented in the study are available from 10.6084/m9.figshare.30913907.

**Funding:** The author(s) received no specific funding for this work.

**Competing interests:** The authors have declared that no competing interests exist.

(LMICs), including sub-Saharan Africa, the health impacts of alcohol and tobacco are increasingly evident as behavioral transitions accompany rapid urbanization, globalization, and demographic change [3–5].

In many African contexts, including Uganda, national health priorities have historically focused on communicable diseases. However, as the epidemiological landscape shifts, there is growing recognition of the dual burden posed by traditional health threats and rising levels of NCDs such as HIV/AIDS and malaria, many of which are driven by modifiable behavioral risk factors such as poor diet, physical inactivity, tobacco use, and harmful alcohol use. Despite this, population-based evidence on the patterns, correlates, and co-occurrence of alcohol and tobacco use in these settings remains limited. Existing research often treats these behaviors in isolation, failing to capture their potential interdependencies and shared determinants [6,7].

Substance use is influenced by a complex interplay of sociodemographic, economic, and health-related factors [7,8]. Prior studies have shown that men, younger adults, individuals with lower socioeconomic status, and those residing in urban or economically deprived settings are more likely to use tobacco or alcohol. However, most of these findings are derived from high-income countries and may not generalize to the sociocultural and policy environments of sub-Saharan Africa. Moreover, few studies have explored the joint modeling of alcohol and tobacco use to account for the potential correlation between these behaviors within individuals [9,10]. This methodological gap limits our understanding of how risk factors may simultaneously influence multiple forms of substance use [11,12].

To address these limitations, this study uses nationally representative survey data to examine the individual and shared correlates of current alcohol and tobacco use in Uganda. By applying both separate generalized linear mixed models and a joint model with a shared random effect, we aim to: (1) identify key sociodemographic and health-related predictors of each behavior; (2) assess the presence of shared latent structures that influence both outcomes; and (3) determine whether joint modeling improves statistical efficiency and model fit compared to separate models. This approach offers a more comprehensive understanding of substance use patterns in Uganda by identifying both unique and overlapping risk factors, thereby providing actionable insights for more integrated prevention strategies.

## Materials and methods

### Study design and data source

This study is a secondary data analysis of the 2014 Uganda WHO STEPwise approach to Surveillance (STEPS) survey, a nationally representative, cross-sectional household survey conducted to assess non-communicable disease (NCD) risk factors. The WHO STEPS methodology provides standardized procedures for data collection, allowing for international comparability and national monitoring [13–15]. The survey covered Ugandan adults aged 18–69 years and collected self-reported and clinical data on behavioral and biological NCD risk factors, including tobacco use, alcohol consumption, physical activity, diet, blood pressure, body mass index, and biochemical markers.

## Study population

The target population was Ugandan adults aged 18–69 years. The analysis included respondents from the 2014 WHO STEPS survey, which was designed to be nationally representative. For this analysis, we included only individuals with complete data on both outcome variables: alcohol consumption and tobacco use. Respondents with missing or invalid responses on these variables were excluded from the analysis [15].

## Ethical statement

We analyzed the 2014 Uganda STEPS NCD Risk Factors Survey data which is publicly available at https://extranet.who.int/ncdsmicrodata/index.php/catalog/633/. No ethical approval was needed as the data were de-identified and ethical approval was obtained in the past from Saint Francis Hospital, Nsambya Institutional Review Board in 2006 and renewed in 2013. The Uganda Ministry of Health NCDs Desk provided the dataset for this analysis following our request.

## Variable definition and coding

**Outcome variables.** The two primary outcomes in this study were alcohol consumption and tobacco use, both coded as binary variables where a response of "Yes" was coded as 1 and "No" was coded as 0. These variables were derived from self-reported survey responses [14] and served as the dependent variables in both the joint and separate logistic regression models.

**Predictor variables.** The predictor variables included demographic, socioeconomic, and behavioral factors. Demographic covariates comprised sex, age group (18–29, 30–44, 45–59, 60–69), and region of residence (Central, Eastern, Northern, and Western). Socioeconomic characteristics included education attainment (no formal schooling, primary, secondary, tertiary/university), marital status (never married, married/cohabiting, separated/divorced/widowed), employment status (unemployed, employed, unpaid/informal), and monthly household income. Health-related variables included body mass index (BMI) categories (underweight, normal weight, overweight, obese), hypertension status (normotensive or hypertensive), and the presence of central obesity. Some covariates, such as education attainment and BMI group, were treated as ordinal variables, while others were categorical or binary. The variable region captured geographic diversity across Uganda, while monthly income provided an economic gradient based on self-reported income.

Table 1 presents the descriptive statistics for the study variables. The majority of respondents were women (60.44%) and most participants resided in the Eastern (29.61%) and Western (31.44%) regions of Uganda. Regarding educational attainment, 40.82% had completed primary school, while only 9.25% had attained tertiary or university education. Most individuals were married or cohabiting (66.86%) and employed (64.77%).

In terms of nutritional status, 63.35% had a normal body mass index (BMI), whereas 13.38% were classified as obese and 15.00% as overweight. Age distribution showed that 40.59% were aged 18–29 years, followed by 35.15% in the 30–44 year category. A majority were normotensive (72.04%), and 35.13% had central obesity. While 63.35% of participants had a normal BMI, 35.13% were classified as centrally obese. This discrepancy reflects the fact that central obesity, measured by waist circumference, can identify individuals with high abdominal fat even when overall body weight (BMI) falls within the normal range.

Behaviorally, 47.22% of the participants reported consuming alcohol, while only 8.56% reported tobacco use. These statistics provide the foundational context for the joint modelling of alcohol and tobacco use explored in this study.

To ensure nationally representative estimates in descriptive analysis, we applied survey weights provided in the WHO STEPS dataset. These weights account for the complex sampling design, including probability of selection, non-response, and post-stratification adjustments. Descriptive statistics, including the prevalence of alcohol and tobacco use, as well as distributions across demographic and health-related variables, were computed using weighted estimates. We used the survey package in R, creating a survey design object with the svydesign() function and estimating proportions using svymean(). This approach aligns with WHO guidelines for analyzing STEPS data and ensures that summary statistics

**Table 1. Descriptive Statistics of the Study Variables.**

| Variable | Level | Frequency | Percent |
|---|---|---|---|
| Sex | Men | 1535 | 39.56 |
| | Women | 2345 | 60.44 |
| Region of residence | Central | 792 | 20.41 |
| | East | 1149 | 29.61 |
| | Northern | 718 | 18.51 |
| | Western | 1220 | 31.44 |
| Educational attainment | No formal schooling | 651 | 16.78 |
| | Primary school | 1584 | 40.82 |
| | Secondary | 1286 | 33.14 |
| | Tertiary/University | 359 | 9.25 |
| Marital status | Married/Cohabitating | 2594 | 66.86 |
| | Never married | 598 | 15.41 |
| | Separated/Divorced/Widowed | 688 | 17.73 |
| Employment status | Employed | 2513 | 64.77 |
| | Unemployed | 412 | 10.62 |
| | Unpaid/Informal | 955 | 24.61 |
| Body Mass Index (BMI) | Normal weight | 2458 | 63.35 |
| | Obese | 519 | 13.38 |
| | Overweight | 582 | 15 |
| | Underweight | 321 | 8.27 |
| Age group | 18-29 | 1575 | 40.59 |
| | 30-44 | 1364 | 35.15 |
| | 45-59 | 678 | 17.47 |
| | 60-69 | 263 | 6.78 |
| Hypertension status | Hypertensive | 1085 | 27.96 |
| | Normotensive | 2795 | 72.04 |
| Central obesity | Normal | 2517 | 64.87 |
| | Obese | 1363 | 35.13 |
| Current alcohol use | No | 2048 | 52.78 |
| | Yes | 1832 | 47.22 |
| Current tobacco use | No | 3548 | 91.44 |
| | Yes | 332 | 8.56 |

presented in Table 1 more accurately reflect the adult population of Uganda aged 18–69 years. Weighted modeling was not feasible due to limitations of multilevel modeling software; however, the complex design was addressed via inclusion of region and sociodemographic covariates.

## Data transformation and modeling approach

To enable joint modeling of the two outcomes, the dataset was reshaped from wide format (one row per individual) to long format (two rows per individual, one for each outcome), using the pivot_longer() function from the tidyr package [16]. The resulting dataset included a binary outcome variable (response) and an outcome type indicator, with a unique identifier for each respondent. After reshaping, the dataset contained N × 2 rows, where N is the number of individuals with complete data on both outcomes.

## Statistical analysis

**Joint model.** To account for the correlated nature of the two outcomes — alcohol consumption and tobacco use — within individuals, we applied a multilevel logistic regression model (also known as a generalized linear mixed-effects model) using the glmmTMB package in R [17]. This approach allows for simultaneous modeling of both outcomes while accounting for unobserved heterogeneity at the individual level.

The model included; Fixed effects for all covariates ($\beta_k$), representing the average effect of each predictor across all individuals, Interaction terms between covariates and the outcome type ($\gamma_l$), allowing covariate effects to vary between alcohol and tobacco use, and a shared random intercept ($u_j$) for each individual ($j$), assumed to follow a normal distribution $u_j \sim N(0, \sigma^2)$. This term captures within-person correlation between the two outcomes and accounts for latent traits influencing both behaviors.

Mathematically, the model is expressed as:

$$\text{logit}\left(P(Y_{ij} = 1)\right) = \beta_0 + \sum_k \beta_k X_{jk} + \sum_l \gamma_l \left(X_{jk} \times \text{outcome\_type}_i\right) + u_j$$

(1)

$$u_j \sim \mathcal{N}(0, \sigma^2)$$

where

- $Y_{ij}$ is the binary response (alcohol or tobacco use) for individual $j$ and outcome $i$,

- $X_{jk}$ are covariates for individual $j$,

- outcome\_type$_i$ indicates the outcome being modeled (alcohol vs. tobacco use),

- $u_j$ is the individual-level random intercept accounting for correlated outcomes within each person.

**Separate models.** To benchmark the joint model, two separate logistic regression models were fitted for each outcome individually (alcohol and smoking), using the same covariates and main effects structure, but without random effects. These were also estimated using glmmTMB [17].

**Model evaluation and comparison.** Model fit was evaluated using the Akaike Information Criterion (AIC) and Degrees of Freedom (DF), which are standard metrics for model selection and comparison [18]. To assess whether modeling alcohol consumption and tobacco use jointly offered advantages over separate models, we computed the AIC for three cases: the joint model, the alcohol-only model, and the smoking-only model. To directly compare the separate models with the joint model, a combined AIC was calculated by summing the individual AIC values of the alcohol and smoking models. This approach is appropriate when evaluating models fitted to different but related outcomes on the same dataset [19]. The same principle was applied to the degrees of freedom (DF), which were also summed to enable a valid comparison with the joint model's DF [20]. This method allows a rigorous assessment of whether the joint model offers a more parsimonious and informative fit by accounting for potential correlation between the outcomes, compared to treating them as entirely separate.

**Visualization of model estimates.** Forest plots were used to visually represent adjusted odds ratios (ORs) and their 95% confidence intervals (CIs) from the joint generalized linear mixed model (GLMM). The GLMM included both fixed and random effects, capturing individual-level heterogeneity across two binary outcomes: current alcohol consumption and tobacco use.

After fitting the joint model using the glmmTMB package, model coefficients were extracted using the broom.mixed::-tidy() function with exponentiation to obtain ORs. The estimates were then filtered to distinguish between:

Main effects: Interpreted as predictors of current alcohol use (reference category of the outcome variable).

Interaction effects: Captured differential effects for tobacco use through interaction terms.

Two separate forest plots were generated using ggplot2; Alcohol Model Effects which shows adjusted odds ratios for all sociodemographic and clinical predictors associated with current alcohol use and smoking-specific Interaction Effects which shows how the effect of each covariate differs for tobacco use compared to alcohol use, interpreted from the interaction terms of the joint model.

Odds ratios and CIs were plotted on a log scale for interpretability. Only statistically significant terms (p < 0.05) were displayed to enhance interpretability and reduce visual complexity. This visual approach allowed clearer comparison of how predictors relate differently to alcohol and smoking behavior when modeled jointly.

### Software and reproducibility

All analyses were conducted using **R version 4.4.3** [21]. The following packages were used; glmmTMB for model estimation [17], readxl for data import, dplyr, tidyr [16], forcats for data manipulation and preprocessing [22].

The full R script used for data preparation and modeling is available upon reasonable request.

## Results

### Model comparison and goodness-of-fit

To assess the appropriateness of modeling alcohol consumption and tobacco use jointly, we compared the fit of a multivariate (joint) logistic regression model with that of two separate univariate models. All models included the same set of covariates reflecting demographic, socioeconomic, and health-related characteristics. The Akaike Information Criterion (AIC) was used to evaluate model fit, with lower AIC values indicating better performance. Table 2 presents the AIC values, degrees of freedom (df), and comparison across the joint model, the separate models for alcohol and tobacco use, and their combined AIC.

The joint model yielded an AIC of 6850.13 with 41 degrees of freedom, which is lower than the combined AIC of 6890.73 from the two separate models (alcohol and smoking combined). This suggests that joint modeling provides a more efficient and parsimonious fit to the data, despite a slightly larger number of parameters.

The lower AIC in the joint model indicates better model fit, suggesting that accounting for within-individual correlation between alcohol and tobacco use improves explanatory power. This result justifies the use of a multivariate mixed-effects logistic regression model and highlights the benefit of jointly analyzing correlated binary outcomes.

### Multivariable analyses of smoking and alcohol use determinants

As discussed earlier, a joint multivariable logistic regression model was fitted to simultaneously examine the associations between individual-level factors and two binary outcomes: current alcohol use and current tobacco use. The model incorporated a random intercept at the individual level to account for within-person correlation between the two outcomes. The model included interaction terms between outcome type (alcohol vs. tobacco) and all covariates, allowing differential

**Table 2. Model Fit Comparison Based on AIC.**

| Model | AIC | Degrees of Freedom (df) |
|---|---|---|
| Joint model | 6850.13 | 41 |
| Alcohol-only model | 5067.99 | 16 |
| Smoking-only model | 1822.74 | 20 |
| Combined separate models | 6890.73 | 36 (16 + 20) |

effects of predictors on each outcome. The random intercept variance (0.77; SD = 0.88) suggested the presence of significant correlation between the outcomes at the individual level.

Table 3 presents adjusted odds ratios (ORs) and 95% confidence intervals (CIs) for associations between sociodemographic and health-related covariates with alcohol and tobacco use, based on three models: separate logistic regressions for smoking and alcohol use, and a joint model with a shared random effect.

In the smoking-only model, female sex was strongly associated with significantly lower odds of smoking (OR = 0.10, 95% CI: 0.07–0.13), while increasing age was associated with substantially higher odds. Notably, adults aged 60–69 had five times higher odds of smoking compared to those aged 18–29 (OR = 5.19, 95% CI: 3.16–8.52). Education level showed a strong inverse association with smoking; tertiary education was associated with a 67% lower odds of smoking. Marital status also played a role, with separated/divorced/widowed individuals more likely to smoke. Interestingly, underweight individuals had over two times higher odds of smoking than those with normal BMI.

In the alcohol-only model, sex differences were also prominent: women had 62% lower odds of alcohol consumption compared to men (OR = 0.38, 95% CI: 0.32–0.44). The association with age remained significant but with more moderate increases. Higher education was associated with lower odds of alcohol consumption, particularly among those with secondary education. Other variables, such as employment status and BMI, did not show significant associations.

The joint model provided a comprehensive view by modeling the two outcomes simultaneously. It confirmed the robust inverse association between female sex and both behaviors (OR = 0.29 for women vs. men). Age gradients persisted for both behaviors, and the effects were generally consistent across smoking and alcohol use. Importantly, the joint model included interaction terms to evaluate differential effects specific to smoking. For example, women had even lower odds

**Table 3. Results from Separate and Joint Models for Alcohol and Tobacco Use.**

| Variable | Smoking Model OR (CI) | Joint Model: Smoking OR (CI) | Alcohol Model OR (CI) | Joint Model: Alcohol OR (CI) |
|---|---|---|---|---|
| Sex (Women vs Men) | 0.10 (0.07, 0.13) | 0.03 [0.01, 0.06] | 0.38 (0.32, 0.44) | 0.29 [0.24, 0.36] |
| Age 30–44 (vs 18–29) | 2.91 (2.02, 4.20) | 1.45 [1.19, 1.76] | 1.46 (1.27, 1.68) | 1.45 [1.19, 1.76] |
| Age 45–59 | 4.22 (2.89, 6.18) | 1.96 [1.53, 2.52] | 2.00 (1.67, 2.39) | 1.96 [1.53, 2.52] |
| Age 60–69 | 5.19 (3.16, 8.52) | 2.57 [1.78, 3.70] | 2.72 (2.02, 3.67) | 2.57 [1.78, 3.70] |
| Region: East | 1.16 (0.77, 1.76) | 0.76 [0.60, 0.96] | 0.77 (0.63, 0.94) | 0.76 [0.60, 0.96] |
| Region: Northern | 2.02 (1.31, 3.12) | 1.13 [0.87, 1.47] | 1.09 (0.87, 1.37) | 1.13 [0.87, 1.47] |
| Region: Western | 1.72 (1.18, 2.50) | 0.83 [0.66, 1.03] | 0.83 (0.68, 1.00) | 0.83 [0.66, 1.03] |
| Education: Primary school | 0.82 (0.59, 1.14) | 0.77 [0.61, 0.97] | 0.81 (0.66, 0.99) | 0.77 [0.61, 0.97] |
| Education: Secondary | 0.43 (0.29, 0.65) | 0.54 [0.42, 0.70] | 0.58 (0.47, 0.72) | 0.54 [0.42, 0.70] |
| Education: Tertiary/University | 0.33 (0.18, 0.63) | 0.81 [0.58, 1.15] | 0.80 (0.61, 1.05) | 0.81 [0.58, 1.15] |
| Marital Status: Never married | 1.33 (0.85, 2.09) | 0.86 [0.66, 1.11] | 1.22 (0.80, 2.04) | 0.86 [0.66, 1.11] |
| Marital Status: Separated/Div/Widowed | 2.21 (1.62, 3.02) | 1.45 [1.16, 1.81] | 0.90 (0.74, 1.12) | 1.45 [1.16, 1.81] |
| Employment: Unemployed | 0.63 (0.37, 1.08) | 0.93 [0.70, 1.23] | 0.90 (0.72, 1.13) | 0.93 [0.70, 1.23] |
| Employment: Unpaid/Informal | 0.77 (0.54, 1.10) | 0.96 [0.78, 1.18] | 0.95 (0.80, 1.14) | 0.96 [0.78, 1.18] |
| BMI: Obese | 0.71 (0.39, 1.27) | 1.02 [0.79, 1.31] | 1.04 (0.85, 1.27) | 1.02 [0.79, 1.31] |
| BMI: Overweight | 0.72 (0.47, 1.10) | 0.93 [0.74, 1.16] | 0.94 (0.78, 1.13) | 0.93 [0.74, 1.16] |
| BMI: Underweight | 2.23 (1.60, 3.10) | 0.99 [0.74, 1.32] | 0.99 (0.77, 1.28) | 0.99 [0.74, 1.32] |
| Hypertension: Normotensive | 1.07 (0.81, 1.42) | 0.92 [0.77, 1.09] | 0.92 (0.79, 1.07) | 0.92 [0.77, 1.09] |
| Central Obesity: Obese | 1.05 (0.71, 1.56) | 1.03 [0.85, 1.26] | 1.03 (0.86, 1.21) | 1.03 [0.85, 1.26] |

***In the joint model, odds ratios for alcohol correspond to main effects, while odds ratios for smoking are obtained by multiplying the alcohol main effect by the smoking-specific interaction term for each covariate.

of smoking in the joint model when this interaction was included (interaction OR = 0.30, 95% CI: 0.19–0.46). Underweight individuals also had significantly higher odds of smoking specifically (interaction OR = 2.45, 95% CI: 1.55–3.88).

Individuals with tertiary education were significantly less likely to smoke (interaction OR = 0.38, 95% CI: 0.19–0.77), but this association was not as pronounced for alcohol use. Several region-specific effects were evident for smoking but not for alcohol consumption, particularly for residents in Northern and Western regions.

Overall, the joint model supports the hypothesis that tobacco and alcohol use share some common predictors, while also exhibiting distinct patterns of association. The presence of significant interaction terms in the joint model highlights the nuanced differences in how demographic and socioeconomic variables influence each behavior. These findings underscore the importance of integrated prevention strategies that address the shared and unique risk factors of both tobacco and alcohol use within populations.

## Associations from the joint model: Forest plot interpretation

The forest plots (Figs 1–3) visually summarize the adjusted odds ratios (ORs) and 95% confidence intervals (CIs) from the joint and separate models for alcohol and tobacco use. Fig 1 presents the results from the joint model, illustrating how factors such as sex, age, education, and region jointly influence the odds of alcohol and smoking outcomes. Notably, Fig 1 shows a strong inverse association between female sex and both outcomes, with amplified effects for smoking (Smoking × Female). Age gradients are also pronounced, with increasing age groups showing higher odds across both outcomes, especially in smoking-specific interactions.

Fig 2 focuses exclusively on the alcohol model and confirms similar trends for sex and age, but with comparatively attenuated estimates and narrower confidence intervals. This plot highlights a strong protective effect of higher education and a regional pattern where Eastern residence is associated with lower alcohol use odds.

Fig 3 represents interaction terms for smoking extracted from the joint model. It estimates and illustrates a sharper gradient in age and sex effects compared to alcohol. It further underscores the relevance of marital status and BMI, with underweight individuals having substantially higher odds of smoking. Together, these figures (Figs 1–3) support the presence of both shared and distinct predictors of alcohol and tobacco use, reinforcing the added value of the joint modeling approach.

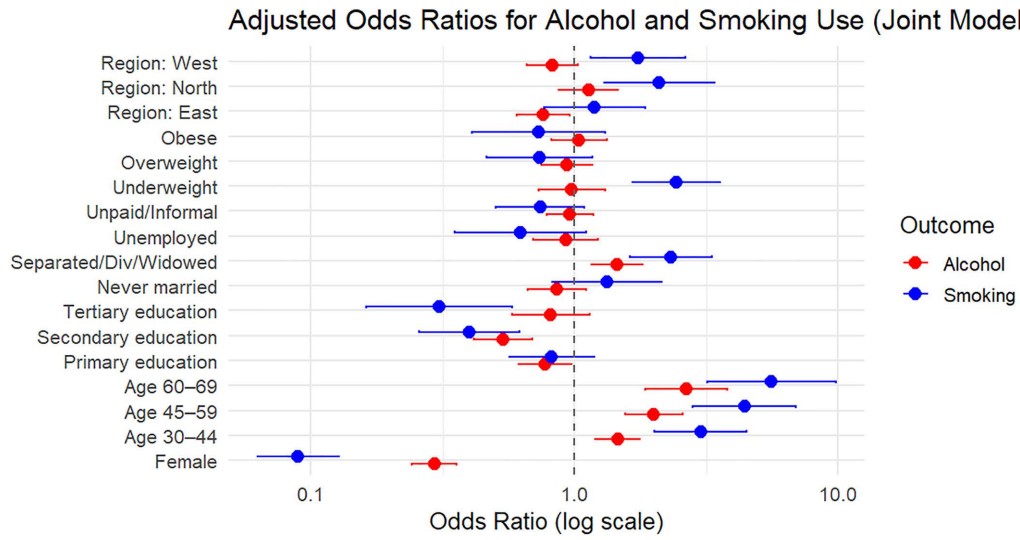

**Fig 1. Adjusted Odds Ratios (ORs) and 95% Confidence Intervals for Alcohol and Smoking Use from the Joint Model.**

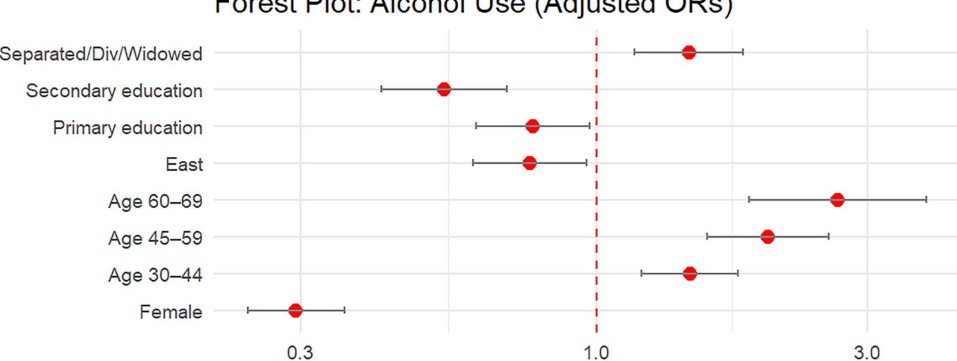

**Fig 2. Forest plot of Adjusted Odds Ratios from Alcohol Model.**

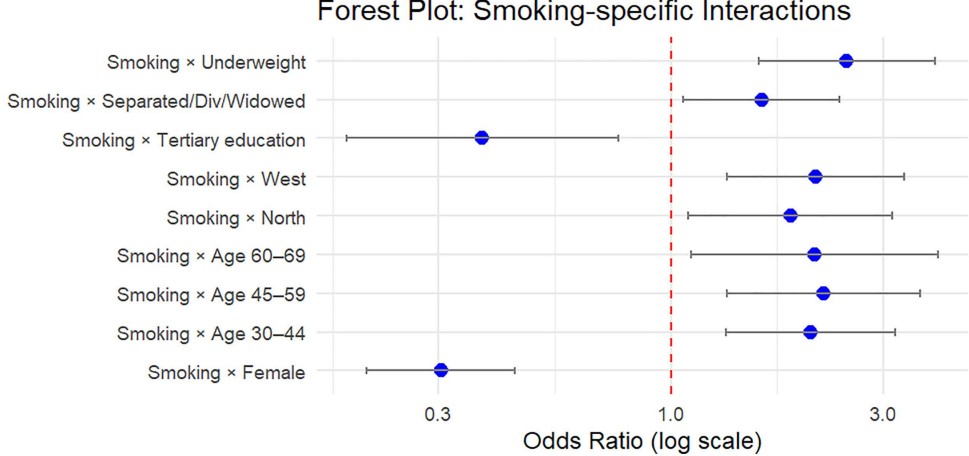

**Fig 3. Smoking-Specific Interaction Effects Derived from the Joint Model.**

## Discussion

This study explored the individual and joint correlates of current alcohol and tobacco use in a nationally representative adult sample. Using separate generalized linear mixed models and a three-level joint model, we identified a range of sociodemographic and health-related factors significantly associated with each behavior. The results revealed both overlapping and distinct predictors, with important implications for integrated behavioral health interventions [4,5].

Sex emerged as a strong and consistent predictor across all models [8]. Women were significantly less likely than men to engage in either alcohol or tobacco use, with the lowest odds observed for smoking. This aligns with existing literature in sub-Saharan Africa, where cultural norms, gender roles, and access to substances may limit female participation in these behaviors [23,24]. However, the interaction terms in the joint model further suggest that the sex gap is more pronounced for smoking than alcohol, underscoring the need for gender-sensitive prevention strategies [8].

Age was positively associated with both alcohol and tobacco use, particularly in older age groups. The strength of the association increased with age in all models, and the interaction terms in the joint model revealed steeper age gradients for smoking than for alcohol. These findings could reflect cohort effects or delayed initiation patterns and suggest the

importance of tailoring interventions to different age strata, especially targeting mid-life and older adults who may have entrenched behaviors [7,25].

Education level was another significant determinant. Compared to individuals with no formal education, those with secondary or tertiary education had significantly lower odds of both alcohol and tobacco use, particularly smoking. The protective effect of education was most prominent in the joint model, where secondary education was associated with a 46% reduction in the odds of the joint outcome. This may reflect greater health literacy and awareness among more educated individuals, as well as socioeconomic factors that influence exposure and access to substances [1,2,6,8,26].

Regional variations were also apparent. Living in Eastern and Western regions was associated with significantly lower odds of alcohol use, while Northern and Western regions showed elevated odds for smoking. These differences likely reflect cultural, economic, and policy-related heterogeneity across regions [8,24]. The joint model confirmed these trends, with interactions indicating that regional effects were more salient for smoking outcomes. This regional heterogeneity suggests the need for locally tailored public health messaging and resource allocation [27].

Marital status and employment status showed nuanced relationships with substance use. Being separated, divorced, or widowed was consistently associated with higher odds of smoking and, to a lesser extent, alcohol use. These associations may reflect psychosocial stressors, loss of social support, or coping mechanisms following relationship dissolution [28,29]. Employment status was not strongly associated with either outcome, although a slight trend toward lower smoking odds was observed among unpaid or informally employed individuals [26,30].

Body Mass Index (BMI) categories and hypertension status had limited associations with alcohol or tobacco use, except for underweight individuals who had notably higher odds of smoking [31]. This may point to bidirectional relationships between smoking and weight loss or poor nutritional status, possibly due to the appetite-suppressing effects of nicotine or comorbid mental health conditions [12].

The joint model provided additional insights by estimating a single latent random intercept to capture shared unobserved heterogeneity between alcohol and tobacco use. The Akaike Information Criterion (AIC) comparison supported the superiority of the joint model over separate models, indicating a more parsimonious fit and highlighting the potential interdependence of the two behaviors [32]. Importantly, interaction terms within the joint model captured differential effects that would be missed in separate models, reinforcing the methodological advantage of joint modeling in behavioral health research [30,33,34].

Taken together, these findings demonstrate that alcohol and tobacco use share several common sociodemographic and contextual correlates, while also exhibiting important differences in their associations with specific factors. The joint modeling results provide evidence of shared underlying influences but do not directly evaluate the effectiveness of integrated prevention or treatment interventions. Future research, particularly longitudinal and intervention-based studies, is needed to assess whether approaches that simultaneously address alcohol and tobacco use lead to improved public health outcomes compared with behavior-specific strategies.

## Conclusion

This study examined the correlates of current alcohol and tobacco use among Kenyan adults using nationally representative survey data and applied both separate and joint generalized linear mixed models. The findings highlight significant associations between substance use and key sociodemographic factors, including sex, age, region, education, and marital status. Notably, women, younger individuals, and those with higher education levels were consistently less likely to report alcohol or tobacco use, while older age, being separated or widowed, and residing in certain regions were linked with increased likelihood of use.

The joint modeling approach revealed a substantial improvement in model fit compared to separate models, providing evidence of a shared unobserved structure influencing both behaviors. This underscores the value of modeling substance use outcomes together when they are likely to be interdependent. The presence of significant interaction terms in the joint model further supports the notion that the determinants of alcohol and tobacco use are not only overlapping but may manifest differently depending on the outcome context.

 

These results have important implications for public health policy and intervention design. Integrated strategies that simultaneously address alcohol and tobacco use, particularly among high-risk demographic groups, may be more effective than fragmented approaches. Programs should also consider the broader social determinants of health—such as education, employment, and regional disparities—when crafting behavior change campaigns or regulatory frameworks.

In conclusion, this study provides a comprehensive and methodologically rigorous assessment of alcohol and tobacco use in Uganda. The results reinforce the need for multi-sectoral, data-driven responses to substance use, particularly as the country continues to confront the dual burden of infectious diseases and rising NCDs. Joint modeling approaches offer an efficient analytical framework for understanding co-occurring risk behaviors and should be adopted more broadly in substance use research and surveillance efforts.

## Supporting information

**S1 Table. Adjusted Odds Ratios and p-values from Reduced Logistic Regression Models for Alcohol and Tobacco Use (Excluding BMI, Hypertension, and Central Obesity).** To assess the robustness of our findings, we conducted a sensitivity analysis excluding potential downstream variables; BMI, hypertension status, and central obesity; from the model. The results were consistent in direction and magnitude with the main models (Supplementary Table S1), suggesting limited overadjustment bias. (DOCX)

**S1 File. Rscript_JointA&T.**
(R)

## Acknowledgments

The authors would like to thank the Uganda Ministry of Health, Non-Communicable Diseases (NCDs) Desk for unconditionally supporting this study by granting data access and providing administrative clearance.

## Author contributions

**Conceptualization:** Grace Kakaire.

**Data curation:** Grace Kakaire.

**Formal analysis:** Grace Kakaire.

**Investigation:** Grace Kakaire.

**Methodology:** Grace Kakaire, Edna Chepkemoi Chumoh.

**Resources:** Grace Kakaire.

**Software:** Grace Kakaire.

**Validation:** Grace Kakaire.

**Visualization:** Grace Kakaire.

**Writing – original draft:** Grace Kakaire, Edna Chepkemoi Chumoh.

**Writing – review & editing:** Edna Chepkemoi Chumoh.

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
