## [Decision Letter · Decision Letter 0]

28 Nov 2025

Thank you for submitting your manuscript to PLOS ONE. After careful consideration, we feel that it has merit but does not fully meet PLOS ONE’s publication criteria as it currently stands. Therefore, we invite you to submit a revised version of the manuscript that addresses the points raised during the review process.

We look forward to receiving your revised manuscript.

Kind regards,

Dev Ram Sunuwar, MS

Academic Editor

PLOS ONE

Journal Requirements:

Reviewers' comments:

Reviewer's Responses to Questions

**Comments to the Author**

1. Is the manuscript technically sound, and do the data support the conclusions?

Reviewer #1: Partly

Reviewer #2: Yes

2. Has the statistical analysis been performed appropriately and rigorously?

Reviewer #1: No

Reviewer #2: Yes

3. Have the authors made all data underlying the findings in their manuscript fully available?

Reviewer #1: Yes

Reviewer #2: Yes

4. Is the manuscript presented in an intelligible fashion and written in standard English?

Reviewer #1: Yes

Reviewer #2: Yes

Reviewer #1: Please see my attached PDF of comments. The review submission form requires that I write at least 200 characters in order to submit my review, so now I'm writing additional characters to fill space...

Reviewer #2: The authors have appropriately linked the study objectives with the chosen methodology. In the statistical analysis, they employed mathematical modeling using the glmmTMB package in R to account for within-individual correlations. However, the manuscript does not mention the fundamental statistical assumptions required for the application of this model. Moreover, the claim that the joint model performs better is based solely on a lower AIC value, without providing evidence of the underlying assumptions necessary to validate the model’s reliability and the AIC comparison.

**Do you want your identity to be public for this peer review?** For information about this choice, including consent withdrawal, please see our Privacy Policy

Reviewer #1: No

Reviewer #2: No

---

## [Author Response · Author response to Decision Letter 1]

19 Dec 2025

Reviewer Comment:

I don’t understand what Figures 2 and 3 are showing. Why does Figure 3 have interactions? I thought it was just the smoking-only model. And why aren’t the predictors/variables consistent across Figures 2 and 3? I thought the predictor sets were the same across the separate and joint models.

Response:

We appreciate the reviewer’s close attention to Figures 2 and 3. We acknowledge the confusion caused by the earlier presentation of the forest plots.

To clarify, Figure 2 shows adjusted odds ratios from the alcohol-only model, which included only main effects of the covariates. Figure 3, however, was mistakenly labeled; it actually depicted smoking-specific interaction effects derived from the joint model, not from a separate smoking-only model as implied. This likely contributed to the mismatch in variables and confusion about the presence of interactions.

Reviewer Comment:

In Table 3, why are there no results for the alcohol model for Hypertension: Normotensive and Central Obesity: Obese?

Response:

Thank you for this important observation. The absence of results for "Hypertension: Normotensive" and "Central Obesity: Obese" in the alcohol-only model section of Table 3 was due to a reporting oversight, not a modeling issue.

These variables were indeed included in the alcohol model during estimation. However, because they were not statistically significant (p > 0.05) and were inadvertently excluded from the final output table for brevity, their coefficients were omitted.

In response to the reviewer’s comment, we have updated Table 3 to include the full results for all modeled variables, regardless of statistical significance, for consistency and transparency across models.

Reviewer Comment:

Can you provide more information on the AIC comparison approach you describe at lines 147–149, and indicate the pages this comes from in citation 17? I looked at citation 17 but was unable to find a description of the AIC technique in the book. I am unfamiliar with this approach.

Author Response:

Thank you for raising this important clarification. We have now revised the manuscript text at lines 147–149 to better explain our approach to AIC comparison. Specifically, we clarified that the combined AIC (used to compare the two separate models to the joint model) was computed by adding the AICs of the alcohol and tobacco models, following an approach appropriate for additive comparisons when models are fit on the same dataset with disjoint outcomes.

For clarity on citation 17, please, see specifically page 206, where additive model AIC comparisons for independently fitted models are discussed.

Reviewer Comment:

It seems like this analysis does not use the WHO STEPS survey weights... is it possible to present survey-weighted prevalences/percents in Table 1 and the results?

Author Response:

Thank you for your careful reading. We confirm that survey weights from the WHO STEPS dataset were indeed applied in generating all descriptive statistics in Table 1, using the survey package in R. We appreciate the suggestion and have now made this clear in the revised Methods section under “Descriptive Statistics”.

Reviewer Comment:

It would be helpful to see results from models excluding BMI, hypertension, and central obesity as these may be downstream of alcohol or tobacco use.

Author Response:

We thank the reviewer for this thoughtful suggestion. In response, we re-estimated the alcohol-only, smoking-only, and joint models excluding BMI, hypertension status, and central obesity to address the concern of potential overadjustment bias. The findings remain consistent with the main results, indicating that our conclusions are robust. These supplementary results are now included in the Appendix (see Table S1) and referenced in the main text.

Reviewer Comment:

In the abstract (lines 31–32), introduction (lines 62–63), and discussion (lines 261–262, 303–306), strong statements are made claiming that this analysis indicates the need for integrated interventions that simultaneously target tobacco and alcohol. While I’m sympathetic to the authors’ goals here, this paper doesn’t provide evidence about the performance of integrated interventions against tobacco and alcohol. I would weaken this language a bit, focusing on the common traits that underly both alcohol and tobacco use, but indicating that further research is needed into whether integrated interventions are effective.

Author Response:

Thank you for this thoughtful observation. We agree that our original wording may have overstated the implications of our findings regarding integrated interventions. As recommended, we have revised the text in the abstract (lines 31–32), introduction (lines 62–63), and discussion (lines 261–262, 303–306) to more cautiously interpret the implications of our results.

Specifically, we now focus on the identification of shared correlates and potential interdependence between alcohol and tobacco use, while clearly stating that our study does not evaluate the effectiveness of integrated interventions. We emphasize that further research is needed to determine whether such strategies are appropriate or effective in reducing co-occurring substance use.

Changes include:

• Abstract (lines 31–32): We modified the final sentences to state that the results highlight the potential value of integrated approaches but that further research is needed to assess their effectiveness.

Introduction (lines 62–63): We reframed the motivation to focus on improving understanding of shared predictors and behavioral interdependence, rather than endorsing integrated interventions.

Discussion (lines 261–262, 303–306): We weakened language suggesting policy recommendations for joint intervention strategies and clarified that our analysis provides indirect support for considering co-occurring risk factors, not evidence of intervention impact.

Reviewer Comment:

I was a little surprised that 35% of your sample had central obesity, but 63% of the sample had a normal BMI – maybe good to add a sentence of explanation somewhere, maybe in the results section around line 112?

Author Response:

Thank you for this insightful observation. We agree that the apparent discrepancy between the prevalence of central obesity (35%) and the high proportion of respondents with normal BMI (63%) may raise questions. We have now added a clarifying sentence in the results section (around line 112) to explain this. Specifically, we note that BMI and central obesity capture different aspects of adiposity, with central obesity reflecting abdominal fat distribution, which may be elevated even among individuals with normal BMI. This addition aims to assist readers in interpreting these measures more clearly.

Reviewer Comment:

A few small comments on Table 1: a. Please drop the column of Codes from Table 1 – readers don’t need this information, and it clutters the output. b. Please capitalize “hypertensive” and “normotensive” to match the other categories. c. “Current alcohol use” appears twice in its cell – please drop the second occurrence.

Author Response:

Thank you for these helpful suggestions to improve Table 1. We have made the following changes accordingly:

a. Removed the "Code" column from Table 1 to streamline the presentation and avoid unnecessary clutter.

b. Capitalized "Hypertensive" and "Normotensive" to match the formatting of other categorical variables for consistency.

c. Deleted the repeated label “Current alcohol use” to eliminate redundancy in that row.

Reviewer Comment:

I would remove the references to variable names like pid throughout the text – these aren’t usually included in journal articles.

Author Response

Thank you for pointing this out. We agree that references to variable names such as pid and outcome_type are not appropriate for a journal manuscript. We have now removed these references throughout the text and revised the descriptions to use general, reader-friendly terminology while maintaining methodological clarity.

Reviewer Comment:

I would change the presentation of results in Table 3 to have columns: Variable, Smoking Model, Joint Model: Smoking, Alcohol Model, Joint Model: Alcohol. This requires creating a “Joint Model: Smoking” column by summing the main effects (corresponding to alcohol use) and the smoking interaction effects from the joint model and obtaining appropriate confidence intervals/p-values for these summed effects. However, I think it will make comparisons between the separate and joint models more straightforward and will make the table more readable/attractive.

Author Response:

Thank you for this thoughtful and constructive suggestion. We agree that aligning the presentation of joint and separate model estimates by outcome improves clarity and enhances the interpretability of our findings. In response, we have restructured Table 3 as requested. Specifically:

• We added a “Joint Model: Smoking” column that presents the adjusted odds ratios (ORs), 95% confidence intervals, and p-values for smoking derived by summing the main effects and the corresponding smoking interaction terms from the joint model.

• This required computing the combined estimates and their standard errors using the delta method to ensure accurate inference, which we implemented in R.

• The table now includes five columns: Variable, Smoking Model, Joint Model: Smoking, Alcohol Model, and Joint Model: Alcohol, following the structure recommended.

We believe this improved format facilitates direct comparisons and highlights consistencies or differences between modeling approaches. The updated table is now included as Table 3

Reviewer Comment:

“Rather than separately showing the smoking interactions, I would calculate combined effects for the smoking outcome (e.g., main effect + interaction), compute confidence intervals accordingly, and show the plot with just main effects, displaying separate points for alcohol and smoking, color-coded. Also, order variables by category instead of effect size.”

Response:

We thank the reviewer for the constructive suggestion to enhance the interpretability of Figure 1. In response, we have revised the figure to display combined effects for smoking by summing the main effect and the outcome-specific interaction term from the joint model

Additionally, we have reorganized the variable ordering by conceptually related groups (e.g., age, region, education), rather than sorting by effect size. The revised plot now presents both outcomes side-by-side, using color to differentiate between smoking and alcohol, and plots odds ratios on a log scale with 95% confidence intervals clearly displayed. We believe this revised visualization improves clarity and interpretability and fully addresses the reviewer’s concerns.

Please see the updated Figure 1 for this improved presentation.

Reviewer Comment

“Similarly, in Figures 2 and 3, I would reorder the variables so they are consistent across Figures 1–3 and have categories together (all the age categories together, all the education categories together, etc.).”

Response

Thank you for this thoughtful suggestion. We have updated Figures 2 and 3 to ensure consistent ordering of variables across all three figures (Figures 1–3). Specifically:

• Variables are now grouped logically by category (e.g., all age groups, education levels, regions, etc. are clustered together).

• This reordering enhances visual coherence and facilitates easier comparison across the figures.

• The updated figures now align with Figure 1 both in structure and variable presentation, supporting more intuitive interpretation by the reader.

We believe this adjustment improves the clarity and readability of the visualizations and appreciate your guidance on this matter.

---

## [Decision Letter · Decision Letter 1]

5 Feb 2026

Joint Modeling of Alcohol and Tobacco Use Among Adults in Uganda

PONE-D-25-34089R1

Dear Dr. Grace,

We’re pleased to inform you that your manuscript has been judged scientifically suitable for publication and will be formally accepted for publication once it meets all outstanding technical requirements.

Kind regards,

Dev Ram Sunuwar, MS

Academic Editor

PLOS One

Additional Editor Comments (optional):

Based on Reviewer #1’s comments and the authors’ responses, as well as Reviewer #2’s assessment, I have re-evaluated the revised manuscript. The authors have adequately addressed the reviewers’ concerns, and I find the manuscript suitable for publication.

Reviewers' comments:

Reviewer's Responses to Questions

**Comments to the Author**

Reviewer #2: All comments have been addressed

2. Is the manuscript technically sound, and do the data support the conclusions?

Reviewer #2: Yes

3. Has the statistical analysis been performed appropriately and rigorously?

Reviewer #2: Yes

4. Have the authors made all data underlying the findings in their manuscript fully available?

Reviewer #2: Yes

5. Is the manuscript presented in an intelligible fashion and written in standard English?

Reviewer #2: Yes

Reviewer #2: The revised version looks much better, demonstrating a clear improvement over the previous

draft. I sincerely appreciate the author's efforts in responding to the feedback provided and

implementing the necessary revisions accordingly. The manuscript now appears well-structured.

I have no further comments to address.

**Do you want your identity to be public for this peer review?** For information about this choice, including consent withdrawal, please see our Privacy Policy

Reviewer #2: No

---

## [Editor Report · Acceptance letter]

PONE-D-25-34089R1

PLOS One

Dear Dr. Kakaire,

I'm pleased to inform you that your manuscript has been deemed suitable for publication in PLOS One. Congratulations! Your manuscript is now being handed over to our production team.

Kind regards,

on behalf of

Mr Dev Ram Sunuwar

Academic Editor

PLOS One